# A Bayesian approach to combining multiple information sources: Estimating and forecasting childhood obesity in Thailand

**John Bryant**[1,2], **Jongjit Rittirong** [2]*, **Wichai Aekplakorn**[3], **Ladda Mo-suwan**[4], **Pimolpan Nitnara**[2]

**1** Bayesian Demography Limited, Christchurch, New Zealand, **2** Institute for Population and Social Research, Mahidol University, Salaya, Nakhorn Pathom, Thailand, **3** Faculty of Medicine, Ramathibodi Hospital, Mahidol University, Bangkok, Thailand, **4** Department of Paediatrics, Faculty of Medicine, Prince of Songkla University, Hat Yai, Songkhla, Thailand

* jongjit.rit@mahidol.edu

## Abstract

We estimate and forecast childhood obesity by age, sex, region, and urban-rural residence in Thailand, using a Bayesian approach to combining multiple source of information. Our main sources of information are survey data and administrative data, but we also make use of informative prior distributions based on international estimates of obesity trends and on expectations about smoothness. Although the final model is complex, the difficulty of building and understanding the model is reduced by the fact that it is composed of many smaller submodels. For instance, the submodel describing trends in prevalences is specified separately from the submodels describing errors in the data sources. None of our Thai data sources has more than 7 time points. However, by combining multiple data sources, we are able to fit relatively complicated time series models. Our results suggest that obesity prevalence has recently starting rising quickly among Thai teenagers throughout the country, but has been stable among children under 5 years old.

**Data Availability Statement:** All data files are available from https://github.com/johnrbryant/bayescomb and https://github.com/johnrbryant/bayescombwho.

## Introduction

Disaggregated estimates and forecasts of social, economic, and health outcomes can support more equitable and effective public policy. Disaggregated estimates and forecasts can be used to identify groups that are being poorly served, to assess the feasibility of policy targets, to provide evidence on the effectiveness of interventions, and to help with priority-setting [1, 2].

Disaggregated estimates and forecasts do, however, required disaggregated data. Assembling datasets with the required level of detail can be difficult. Household surveys may have the variables needed, but sample sizes are often too small to support the desired level of disaggregation. Population censuses have large samples, but are carried out infrequently. Administrative data or big data, such as tax records or cellphone data, have large samples and high frequency, but often miss parts of the target population, and have substantial measurement errors [3]. A particular problem for disaggregated forecasting is assembling long time series of

**Funding:** JR was responsible for funding acquisition under the project entitled "Childhood overweight and obesity policy research project (grant no. 62-00269)" funded by Thai Health Promotion Foundation (https://en.thaihealth.or.th/). The funders had no role in study design, data collection and analysis, decision to publish, or preparation of the manuscript.

**Competing interests:** The authors have declared that no competing interests exist.

input data. The greater the level of disaggregation, the more frequently geographical boundaries, and definitions of variables and target populations, change [4]. When the time series of input data are short, forecasting is challenging [5, Section 13.7].

One way to meet the demands for disaggregated data is to combine information from many different sources [6, 7]. Ideally, the sources should have complementary strengths and weaknesses, so that gaps in any one source can be filled by others. In many applications, it is necessary to be opportunistic: to design the models around the data that are available. It is, nevertheless, important to base the models on an explicit statistical framework, to properly account for uncertainty [8] and for transparency and replicability.

Bayesian statistical models are particularly well suited to combining multiple sources of information. The key distinction between Bayesian statistics and frequentist statistics is that Bayesians are willing to use probability distributions to represent the state of knowledge about any uncertain quantity [9, 10]. Probabilities act as a common unit of measurement, allowing Bayesians to combine many sources and types of information within the same model. In the final model presented in this paper, for instance, we use probability distributions to represent sampling variability, the plausible range for measurement errors, and the plausible range for annual variation in obesity rates.

In this paper, we use a Bayesian approach to combining information from multiple sources to produce disaggregated estimates and forecasts of childhood obesity in Thailand. The application is an important one. Obesity has emerged as a major health issue throughout the world, including in middle income countries such as Thailand [11, 12]. Disaggregated estimates provide information on the size of the problem, and on risk factors: whether boys are more at risk than girls, for instance, or whether urban children are more at risk than rural children. Disaggregated forecasts help with planning and priority-setting by health agencies by showing the scale of the problem over the coming years.

Our overall approach is to specify a system model describing the underlying rates and data models describing the relationship between the rates and the available data, and then jointly infer all unknown quantities. This type of hierarchical modelling is becoming increasingly common in demography, epidemiology, ecology, and related disciplines [e.g. 13–17], though it is still uncommon in studies of obesity. Distinctive features of our analysis include the diversity of the information sources that we combine, the flexibility of our model of prevalences, the integration of estimation and forecasting, and the important role played by informative prior distributions.

We start the paper with a simple model, and then progressively add extensions. The initial model is restricted to national-level trends, uses a single data source, and has no provision for measurement error. The first extension is to bring in two extra data sources and allow for measurement error. The second extension is to use international estimates of obesity trends to create informative priors, to reduce uncertainty about rates of change. The third extension is to add region and urban-rural residence. The fourth extension is to reformulate the prior describing the accuracy of the administrative data, in response to implausible patterns in the disaggregated results. We conclude the analysis by probing some of our modelling assumptions. In the final section of the paper, we argue that our methods are applicable to a wide range of estimation and forecasting problems. Data and code to replicate the analysis are available at https://github.com/johnrbryant/bayescombwho and https://github.com/johnrbryant/bayescomb.

## Data

### National Health Examination Survey

We use data from the 1991, 1997, 2004, 2009, and 2014 rounds of the Thai National Health Examination Survey (NHES). The survey is nationally representative, with a complex design

including stratification and clustering. Interviews are conducted by local health personnel, who also measure the height and weight of respondents [18, 19]. We restrict our analysis to respondents aged 2–17 years. The number of respondents in the target ages varies from 661 in 2004 to 9,247 in 2009, with an average of 6,058. Some rounds of the survey omit some ages within the 2–17 range.

## 2001 Holistic Development of Thai Children Survey

The 2001 Holistic Development of Thai Children (HDTC) Survey collected data from 17 provinces on topics related to family and childrearing, including heights and weights of children [20]. We use data on height and weight for 2,127 respondents aged 2–4 years.

## Schools data

The Thai Office of Basic Education Commission collects data on the health and socio-economic status of students at preschools, elementary schools, and high schools receiving government subsidies. The data collected includes heights and weights. Sample sizes are large, ranging from 6,380,145 students aged 2–17 in 2019 to 6,883,478 in 2013. Using information on the locations of schools, we construct a variable distinguishing between eight regions of Thailand, and a variable distinguishing urban areas from rural areas.

Coverage of the schools data is incomplete, with the number of children in the dataset representing, on average, only 51% of all children in the corresponding age groups. Coverage is uneven across regions and age groups, though children aged 2–9 in Bangkok have by far the lowest coverage, with coverage rates of less than 5%. Measurement of height and weight is also uneven. Although teachers are supposed to take measurements themselves, anecdotal evidence suggests that in some schools teachers merely ask children their heights and weights. The individual-level weight and height measurements also have some heaping around values ending in 0 and 5.

## Population estimates

To obtain population estimates for the period 1990–2010, we start with counts by age, sex, region, and urban-rural residence from 1% census sample files for 1990, 2000, and 2010 [21], and then interpolate between census years using splines, with one spline for each combination of age, sex, region, and urban-rural residence. For the period 2011–2019, we use 2010-base population projections from the Thai National Economic and Social Development Board [22].

## WHO international obesity estimates

We use WHO annual estimates of obesity prevalence among children aged 5–19 for 191 countries for the period 1990–2016, downloaded from the WHO website [23]. The estimates are derived from many data sources, and involve considerable imputation, interpolation, and smoothing. The estimates distinguish between females and males, but not between age groups. The WHO uses slightly different thresholds for height and weight to define obesity than we do with the Thai data. All WHO estimates come with confidence intervals.

## Ethics statement

This study was approved by the Institute for Population and Social Research Institutional Review Board (IPSR-IRB), at Mahidol University, Thailand (COE. No. 2019/07–278).

## Direct estimates of obesity prevalence

We start with some direct estimates of obesity prevalence that do not depend on statistical modelling. We measure obesity using body mass index (BMI), which is defined as weight in kilograms divided by the square of height in centimeters. A child is classified as obese if the child's BMI exceeds the age-sex-specific cutoffs proposed by [24].

Fig 1 shows direct estimates of prevalence based on the NHES, HDTC, and schools data. We calculate point estimates and 95% confidence intervals from the survey data using standard design-based methods, as implemented in the *R* package **survey** [25]. The schools estimates are simply the number of students who are obese divided by the number of students in the schools dataset.

The estimates in Fig 1 suggest that obesity prevalence is trending upwards in all age groups, with the possible exception of children aged 2–4. There is also evidence that obesity is rising faster among males than among females. The precision of the survey estimates varies across different combinations of year and age, reflecting differences in sample sizes. Schools data yields lower prevalence estimates than survey data, except among the youngest children, though in most cases the trends are in the same direction.

To measure differences between areas within Thailand, we rely entirely on schools data. Subnational prevalence estimates for females are shown in Fig 2. Results for males, which are similar to those for females, are shown in the S1 File. There do appear to be small differences between regions in obesity prevalence. Moreover, with one exception, these differences appear to be consistent across age-sex groups, with regions that have high prevalences in one age-sex group having high prevalences in all the other age-sex groups. The exception is Bangkok, which, according to the schools data, has relatively high obesity below age 10, and relatively low obesity above age 10.

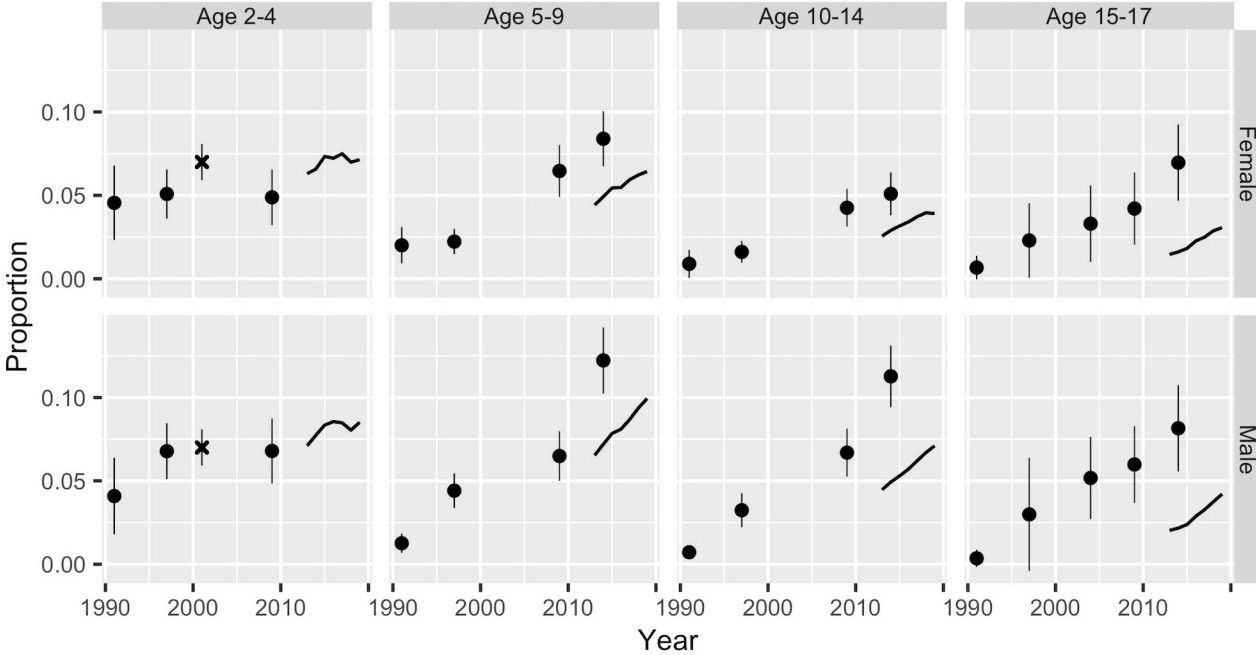

**Fig 1. Direct estimates of national-level obesity prevalence by age, sex, and year from the NHES, HDTC, and schools data.** The dots represent point estimates from the NHES, and the x's represent point estimates from the HDTC; the vertical lines represent the associated 95% confidence intervals. The HDTC data does not distinguish females and males, so the figure shows estimates for both sexes combined. The lines for the period 2013–2019 are prevalence estimates from schools data.

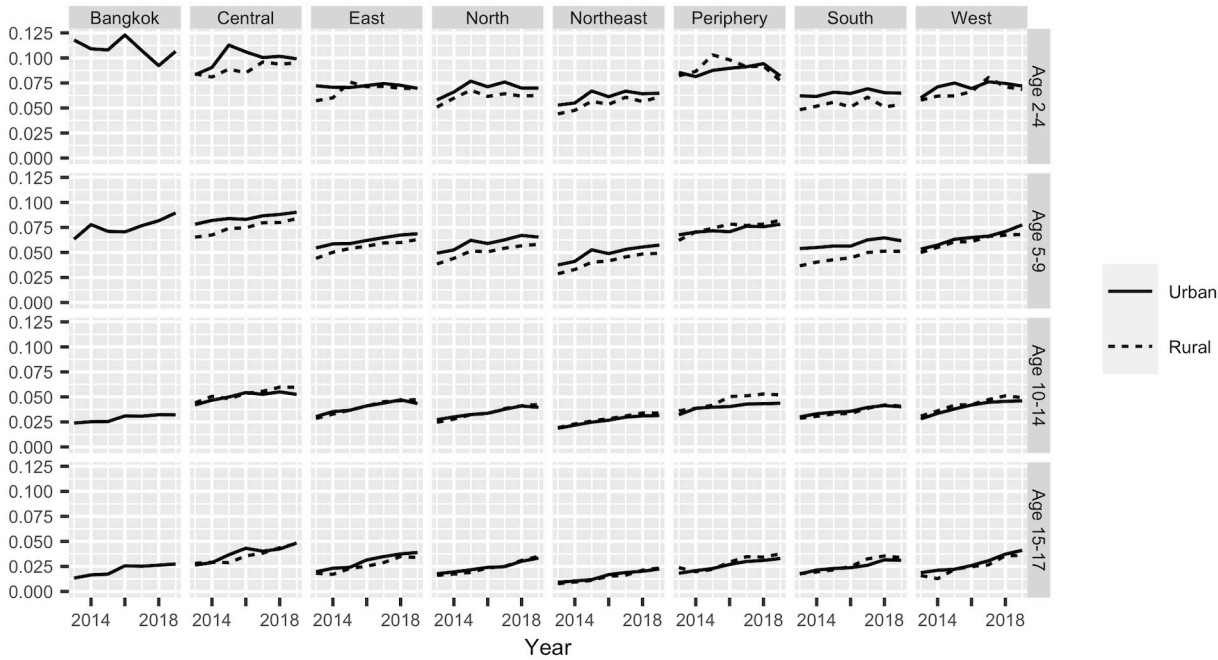

**Fig 2. Direct estimates of obesity prevalence by age, region, and urban-rural residence for females, based on schools data.**

## Model 1: National-level estimates and forecasts, using only NHES data

### Methods

Our first statistical model deals with obesity prevalence, by age and sex, at the national level, using only the NHES to measure obesity. Within each combination of age group $a$, sex $s$, and time $t$, we treat the number of children who are obese, $y$, as a draw from a binomial distribution with sample size $n$ and probability $\pi$. Parameter $\pi$ is the probability that a randomly-chosen child is obese, and is the main quantity that we wish to infer.

Our model assumes that, within each combination of age, sex, and time, respondents are a simple random sample of all Thai children. Respondents in the NHES are not in fact sampled in this way. The complex survey design implies that, even after conditioning on age, sex, and time, some groups of Thai children have different probabilities of being included in the NHES sample than others. Following [26, 27], we account for the non-representativeness of the sample by fitting our model to 'effective' counts of respondents rather than to raw counts. Effective counts equal raw counts scaled by factors that depend on sample weights. An analysis that treats effective counts as if they come from a simple random sample yields approximately the same means and variances as a more complicated analysis that explicitly accounts for the complex survey design. The S1 File gives the details.

Our model relating NHES data to prevalence $\pi_{ast}$ is

$$y_{ast}^{\text{EffN}} \sim \text{Binomial}(\pi_{ast}, n_{ast}^{\text{EffN}}), \tag{1}$$

where the 'EffN' superscript denotes effective counts from the NHES. Prevalence $\pi_{ast}$ is in turn modelled, on a logit scale, as a draw from a normal distribution, the mean of which is the

product of row vector $x_{ast}$ and column vector $\beta$,

$$\text{logit}(\pi_{ast}) \sim \text{N}(x_{ast}\beta, \sigma^2). \tag{2}$$

Transformation to the logit scale implies that values are no longer bounded by 0 and 1. Vector $\beta$ contains an intercept, main effects for age, sex, and time, and interactions between age and sex, age and time, and sex and time. Vector $x_{ast}$, which consists entirely of 1s and 0s, assigns the appropriate main effects and interactions to each combination of age, sex, and time. The main effects and interactions capture demographic regularities. For instance, the age main effect captures the average age pattern across both sexes and all times, and the age-sex interaction captures systematic differences between the age patterns of females and males.

Estimates of the main effects and interactions can be stabilized by adding to the model information about plausible ranges and patterns. With a Bayesian model, this sort of additional information can be encoded in prior distributions.

With the sex effect, we use a relatively simple prior distribution,

$$\beta_s^{\text{sex}} \sim \text{N}(0, 1). \tag{3}$$

This prior captures the idea that, on a logit scale, we might see female-male differences of -0.1 or 1.2, for instance, but not -10 or 120. Bayesians refer to priors like this, which seek only to rule out highly implausible values, rather than providing tight bounds on a parameter, as 'weakly informative' [10, 28].

The prior for the age effect is identical to the prior for the sex effect. The prior for the intercept term has the same form as the prior for the sex effect, but has a standard deviation of 10.

For time, we use a 'damped linear trend' prior [29], which is a flexible version of a random walk with drift,

$$\beta_t^{\text{time}} \sim \text{N}(\alpha_t, \tau_\beta^2) \tag{4}$$

$$\alpha_t \sim \text{N}(\alpha_{t-1} + \delta_{t-1}, \tau_\alpha^2) \tag{5}$$

$$\delta_t \sim \text{N}(\phi\delta_{t-1}, \tau_\delta^2). \tag{6}$$

Time effect $\beta_t^{\text{time}}$ equals a level term $\alpha_t$ plus some random noise, the magnitude of which is governed by $\tau_\beta$. The level term at time $t$ equals its value in time $t-1$, plus some random noise governed by $\tau_\alpha$, plus a drift term $\delta_{t-1}$. The drift term captures any tendency for upward or downward trends in obesity to persist over time. Empirical studies of time series models have found that damping upward or downward trends, rather than allowing them to continue indefinitely, tends to improve forecast accuracy [30, 31]. In our specification, damping is controlled by parameter $\phi$. Parameter $\tau_\delta$ governs the amount of random noise in $\delta_t$.

To complete the prior for the time effect, we need to specify priors for standard deviations $\tau_\beta$, $\tau_\alpha$, and $\tau_\delta$, and for $\phi$. With each of the $\tau$s, we use a half-normal distribution with a standard deviation of 1. A half-normal distribution has the same shape as a normal distribution with mean zero, but limited to non-negative values. We restrict $\phi$ to the range [0.8, 1], and assume that

$$\frac{\phi - 0.8}{1 - 0.8} \sim \text{Beta}(2, 2). \tag{7}$$

All these priors qualify as weakly informative.

As we discuss below, the flexibility of the damped linear trend prior makes it challenging to fit. However, less flexible versions of the prior could potentially miss important features of the

data. It is, for instance, tempting to assume that, within each combination of age and sex, rates of change are constant over time. Doing so would, however, reduce our ability to detect turning points, and could produce forecasts that were inappropriately confident.

The prior for the interaction between age and sex has the same structure as the prior for the age and sex main effects. The prior for the interaction between age and time is a variant on the prior for time, in which each age group has its own linear trend model, but the variance and damping parameters are shared across all age groups. The prior for the interaction between sex and time is a linear trend model. Finally, the $\sigma$ in (2) has the same half-normal prior as the $\tau$s from the prior for time.

We estimate the model using function `estimateModel` from open source *R* package **demest**, available at github.com/statisticsnz/R. Function `estimateModel` uses Markov chain Monte Carlo methods, customised for demographic estimation. The output from the modelling is a sample of values from the joint posterior distribution for all unknowns quantities in the model [28]. We summarise the joint posterior distribution by calculating medians, 50% credible intervals, and 95% credible intervals of the distributions for any unknown quantities we are interested in. The posterior medians serve as point estimates.

The estimation process includes imputing values for years in which there is no data. It is traditional to refer to this imputation process as interpolation when the values being imputed lie in the past, and as forecasting when the values lie in the future. In our model, however, there is no strong distinction between imputation of past values and imputation of future values, with exactly the same specification being used for both.

## Results

The top panel of Fig 3 shows results for prevalence $\pi_{ast}$ from our initial model. Although the median estimates and forecasts, represented by the white lines, look reasonable, the 50% and 95% credible intervals, represented by the dark and light bands, are implausibly wide in years not covered by the NHES. From 2020 onwards, the 95% credible intervals essentially cover the entire range from 0 to 1.

Examining estimates for higher-level parameters (not shown) indicates that most of the uncertainty about prevalence $\pi_{ast}$ comes from uncertainty about time effects, and about age-time and sex-time interactions. The parameter in the priors for time effects and interactions that has the biggest influence on uncertainty is $\tau_\delta$, governing changes in the drift term. Higher values for $\tau_\delta$ imply bigger changes in drift term $\delta_t$, which, since the results compound over time, greatly increases the scope for extreme outcomes.

The $\tau_\delta$ parameters in the priors for the time, age-time, and sex-time terms are estimated imprecisely. The 95% credible interval for the $\tau_\delta$ in the time term, for instance, is (0.004, 0.563). Values towards the upper end of this range permit huge year-on-year changes in $\delta_t$.

The reason that $\tau_\delta$ and the other parameters in the linear trends priors are estimated imprecisely is that, with only five rounds, the NHES provides limited information on change over time. Five time points is large for a health survey, but tiny for a time series model. For instance, of the 100,000 time series used in the M4 time series competition—a major empirical comparison of time series models—the shortest series had 15 time points [31].

## Model 2: Adding HDTC and schools data

### Methods

In our first extension, we expand our model to accommodate HDTC and schools data. Fig 4 compares the structures of our first and second models. Both models deal with counts of obese children, but the nature of these counts differ. In the first model, the counts are of obese

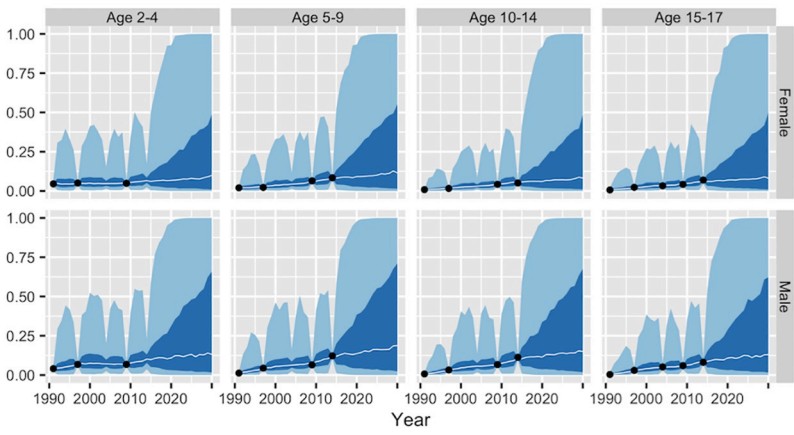

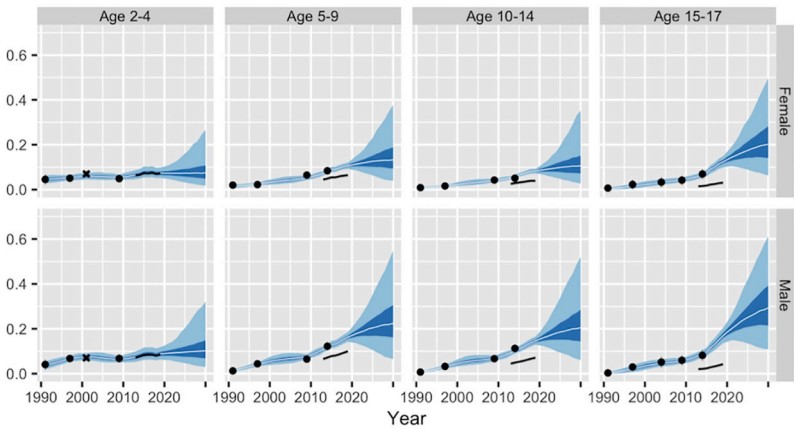

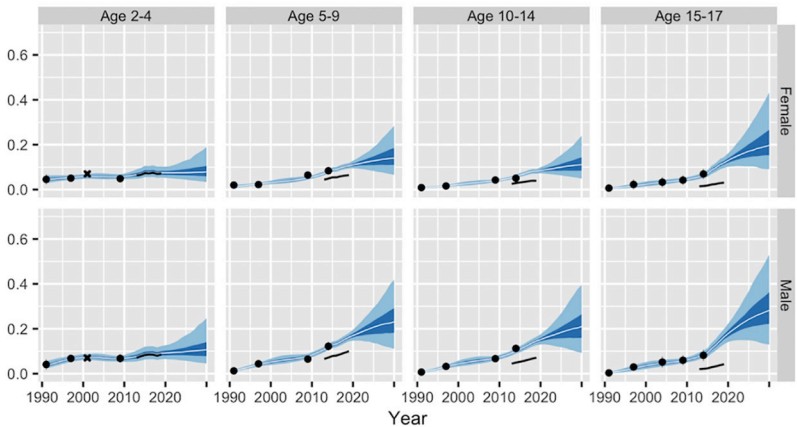

**Fig 3. Estimates and forecasts of national obesity prevalence from three models.** The top panel shows results for Model 1 using NHES data only, the middle panel shows results for Model 2 using NHES, HDTC, and schools data, and the bottom panel shows results for Model 3 with NHES, HDTC, and schools data plus WHO-based prior distributions for time terms. The light bands represent 95% credible intervals, the dark bands represent 50% credible intervals, and the white lines represent medians. The black symbols represent direct estimates from Fig 1. The vertical axis for the top panel extends from 0 to 1, while the vertical axes for the other panels extend from 0 to 0.7.

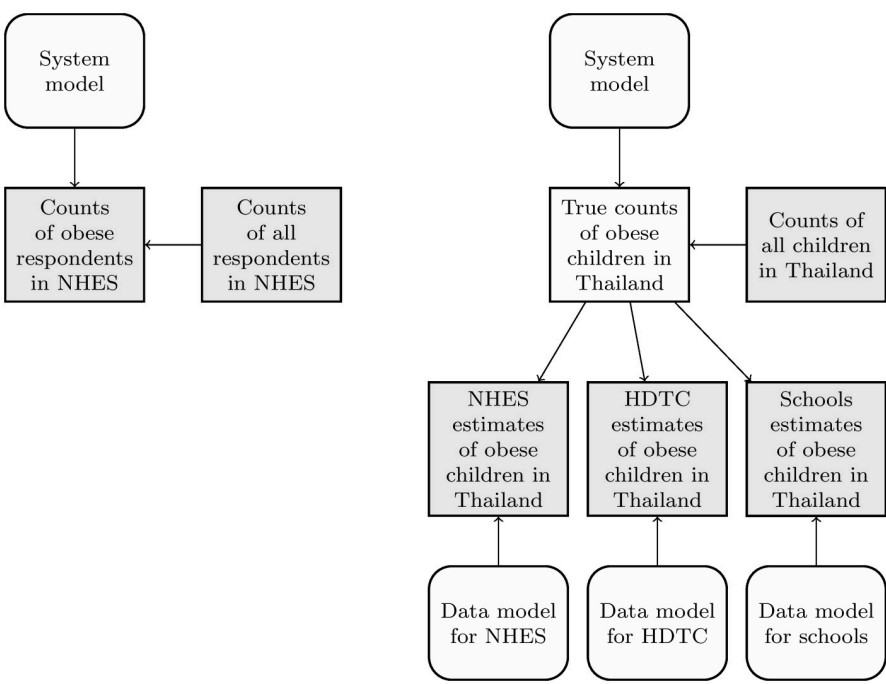

**Fig 4. The structure of our first and second models.** Our first model, on the left, allows for a single data source with sampling errors but not measurement errors. Our second model, on the right, allows for multiple data sources, all with sampling and measurement errors. Observed quantities are shaded gray; everything else is unobserved, and must be inferred.

children in the NHES, which are known with certainty. In the second model, the counts are of obese children in all of Thailand, which are unknown, and must be inferred. Even in the second model, however, total numbers of children at risk of obesity, disaggregated by age and sex, are treated as known.

The counts of obese children in the second model are inferred from three datasets. Each of these datasets provides imperfect and incomplete measurements of obese children in all of Thailand. The NHES dataset in our second model differs from the NHES dataset in our first model. Rather than being a set of effective counts from the NHES survey, it is a set of estimates for children in all of Thailand in each year of the survey. These estimates are obtained from individual-level NHES data using standard methods for complex survey data, as implemented in the **survey** package. Similarly, the HDTC dataset consists of an estimate for ages 2–4 in 2001 constructed from the raw individual-level HDTC data. The schools dataset is constructed by multiplying schools-based prevalence estimates like those in Fig 1 by population estimates for the corresponding age, sex, and year.

The system model for our second overall model replaces (1) from our first overall model with

$$y_{ast}^{\text{True}} \sim \text{Binomial}(\pi_{ast}, n_{ast}^{\text{True}}). \tag{8}$$

In every other way, including all the prior distributions, the system models for Model 1 and Model 2 are identical.

To construct our data model for the NHES, we rely on features of the design of the survey, which is a common strategy in Bayesian analyses of multiple data sources [8, e.g. 15]. The design of the survey implies that the NHES estimates should be unbiased, and that errors in

these estimates should be approximately normally distributed. Moreover, the standard deviations for these errors can be estimated through design-based methods that exploit information about the survey including sample weights.

Our data model for the NHES is

$$y_{ast}^{\text{EstN}} \sim \text{N}(y_{ast}^{\text{True}}, \kappa_{ast}^2), \tag{9}$$

where the superscript 'EstN' denotes 'estimates derived from the NHES.' We set the $\kappa_{ast}$ equal to the standard deviations that the *R* package **survey** produces alongside the estimates $y_{ast}^{\text{True}}$. (Code for the design-based calculations s included in the repository https://github.com/johnrbryant/bayescomb).

Values for $y_{ast}^{\text{EstN}}$ are available for only some values of $y_{ast}^{\text{True}}$. For instance, no values for $y_{ast}^{\text{EstN}}$ are available for ages 2–4 and 5–9 in 2004, or for any age groups in 2005–2008. The gaps in the data pose no difficulties for estimation: the data model uses the values of $y_{ast}^{\text{True}}$ that it needs to predict the corresponding values in the NHES dataset, and ignores the rest.

The data model for the HDTC estimates has the same structure to the NHES model, except that the subscripts change from *ast* to *at*, since the HDTC data does not specify the sexes of the children. The fact that the data lack a dimension contained in $y_{ast}^{\text{True}}$ also does not pose any difficulties for estimation: within the estimation process, the sex dimension in $y_{ast}^{\text{True}}$ is aggregated away before the values are supplied to the data model.

The data model for the schools dataset is

$$\log y_{ast}^{\text{EstS}} \sim \text{N}(\log y_{ast}^{\text{True}} + \gamma_a, \varsigma_y^2) \tag{10}$$

$$\gamma_a \sim \text{N}(0, \varsigma_\gamma^2). \tag{11}$$

The log of the schools-based estimate equals the log of the true count, plus an age-specific bias term $\gamma_a$, plus random noise. The use of logs implies that the data model is expressed in terms of percentage errors, rather than absolute errors. We focus on differences in biases across age groups because comparisons of direct estimates, shown in Fig 1, suggests that these differences are large. As illustrated in Fig 2 in the S1 File, dropping the age-specific bias term from the model produces an implausible jump in obesity prevalences for ages 2–4 when moving from NHES and THDS data to schools data. In principle, we could extend the model to allow for systematic differences across other dimensions besides age. However, allowing too much flexibility in data models can make the overall model difficult to fit.

The age-specific bias terms are drawn from a common distribution centered on 0. Parameter $\varsigma_y$ governs the amount of random noise, and parameter $\varsigma_\gamma$ governs variability in age-specific bias. We use half-normal priors with standard deviations of 1 for $\varsigma_y$ and $\varsigma_\gamma$.

The likelihood from combining the three datasets has the form

$$
\begin{aligned}
p(\text{data}|\text{true obesity counts}) \quad &= p(\text{NHES data}|\text{true obesity counts}) \\
&\times p(\text{HDTC data}|\text{true obesity counts}) \\
&\times p(\text{Schools data}|\text{true obesity counts}).
\end{aligned}
$$

The (unobserved) true obesity counts appear multiple times in the likelihood. The repetition does not, however, cause any problems. It is analogous to having the same regression coefficients occur multiple times in the likelihood for a regression model.

## Results

The middle panel of Fig 3 shows results from Model 2. Adding the extra data sources dramatically reduces uncertainty compared with the first model. (Note that the top and middle panels of Fig 3 use different vertical scales.) The reduction in overall uncertainty results, in large part, from a reduction in uncertainty about the $\tau_\delta$ parameters in the main effect and interactions involving time. For instance, the 95% credible interval for $\tau_\delta$ in the main effect for time in the second model is (0.001, 0.063), compared with (0.004, 0.563) in first model.

With uncertainty at more reasonable levels, differences in prevalences by age and sex become more apparent. The estimates for ages 2–4 imply that prevalences rose only slightly during the 1990s and 2010s. The relatively flat forecasts for ages 2–4 extrapolate these slow changes into the future. The estimates have more pronounced upward trends among other age groups, particularly for males. Prevalences rise particularly quickly among males aged 15–17 during the 2010s and 2020s. These changes appear to reflect the rapid growth (from a low base) in obesity among males aged 15–17 in the schools data.

The results for Model 2 are consistent with the idea that the relationship between actual prevalence and prevalence in the schools data varies with age. School-based prevalences more or less match the modelled estimates at ages 2–4, but are substantially lower among the other age groups.

## Model 3: Adding WHO-based priors for time effects

### Methods

Although our second overall model gives more sensible estimates of uncertainty than the first, the second model still appears to overstate the probability of sudden shifts in prevalences, judging by the wide 95% credible intervals for 2030. This suggests that, even with the addition of the HDTC and schools data, the parts of the model dealing with change over time could benefit from more data. In the absence of more Thai data on change over time, we turn to data for other countries—specifically, the WHO estimates of obesity trends in 191 countries.

Our strategy is to fit a simple model covering all 191 countries, and extract from the model parameter values capturing typical year-to-year variability. We use these parameters to formulate informative priors for parameters governing change over time in our Thai model. A prior is informative if it places relatively tight bounds on a parameter, and thus potentially has a substantial effect on final estimates.

We convert the WHO point estimates and 95% confidence intervals into effective counts of children, as described in the S1 File. We then fit the model

$$y_{cst}^{\text{WHO}} \sim \text{Binomial}(\pi_{cst}^{\text{WHO}}, n_{cst}^{\text{WHO}}) \tag{12}$$

$$\text{logit}(\pi_{cst}^{\text{WHO}}) \sim \text{N}(x_{cst}^{\text{WHO}}\beta^{\text{WHO}}, \sigma_{\text{WHO}}^2), \tag{13}$$

where $c$ denotes country, $s$ sex, and $t$ time. Vector $\beta^{\text{WHO}}$ contains a country effect, a sex effect, a country-sex interaction, and a time effect. The sex effect has a normal prior with a standard deviation of 1. The prior for the country effect is $\beta_c^{\text{ctry}} \sim \text{N}(0, \tau_{\text{ctry}}^2)$, with $\tau_{\text{ctry}}$ having a half-normal prior with standard deviation 1. The prior for the country-sex interaction is identical to the prior for the country effect, except that the $c$ subscript is replaced by a $cs$ subscript. The time effect has a linear trend prior, with exactly the same specification as the prior for time effects and interactions in our Thai models.

Having fitted the model to the WHO data, we discard all results except those for the time effect. In particular, we retain samples from the posterior distributions for time effect parameters $\tau_\beta$, $\tau_\alpha$, $\tau_\delta$, and $\phi$. We use these samples to create informative priors for the corresponding

parameters in the time effect, age-time interaction, and sex-time interaction in our models for Thailand.

Based on the sample from the posterior distribution for $\tau_\beta$ in the WHO model, for instance, it appears that the posterior distribution can be closely approximated by the half-normal distribution $N^+(0, 0.0054^2)$. The distribution $N^+(0, 0.0054^2)$ can be used as an informative prior for $\tau_\beta$ in the time effect, age-time interaction, and sex-time interaction in the Thai model. Informative priors for $\tau_\alpha$, $\tau_\delta$, and $\phi$ in the time effect, age-time interaction, and sex-time interaction can be constructed in similar ways. Details are provided in the S1 File.

An important feature of our use of the WHO data is that we only extract information about rates of change. The parameters $\tau_\beta$, $\tau_\alpha$, $\tau_\delta$, and $\phi$ all deal with year-to-year variation, and not with absolute levels. Even with the WHO-based priors for time effects, our Thai models continue to rely on Thai data to determine absolute levels. We assume that even if differences between the WHO and Thai data in the age groups covered and in the thresholds used to define obesity make it difficult to pool information about absolute levels, they do not affect our ability to pool information about rates of change.

The main advantage of the WHO estimates is that they summarise the experiences of virtually every country in the world. The main disadvantage, for our purposes, is that many country-specific time series have been subject to substantial smoothing, imputation, and interpolation. The resulting time series may understate actual year-to-year variation. To allow for this possibility, we modify the prior distributions for $\tau_\beta$, $\tau_\alpha$, $\tau_\delta$, and $\phi$ so that they have approximately twice the variance of the original WHO-based versions, and use these modified priors, rather than the original informative priors, in models 3–5. (The S1 File includes a description of the modification process.) The decision of how much to scale the variances of the priors is inevitably somewhat arbitrary. We test the sensitivity of our results to alternative choices later in the paper.

## Results

The bottom panel of Fig 3 shows results from Model 3. Comparison of the middle and bottom panels of Fig 3 suggests that the use of WHO-based priors has virtually no effect on historical estimates. Use of the WHO-based priors does, however, lead to somewhat narrower credible intervals for 2030. Exploiting the information about plausible annual variation contained in the WHO estimates leads to a modest increase in the precision of the forecasts.

## Model 4: Disaggregating by region and by urban-rural residence

### Methods

In Model 4, we disaggregate the estimates and forecasts by region and by urbal-rural residence. Eq (8) in the system model is replaced by

$$y_{asrut}^{\text{True}} \sim \text{Binomial}(\pi_{asrut}, n_{asrut}^{\text{True}}), \tag{14}$$

which is identical to (8), except for the addition of the $r$ subscript denoting region and the $u$ subscript denoting urban-rural residence.

The main effects used to predict $\text{logit}(\pi_{asrut})$ in Model 4 are the same as those in Model 3, along with a main effect for region and a main effect for urban-rural residence. The prior distributions for region and urban-rural residence have the same format as the prior distribution for country in the WHO model.

The data models for the NHES and HDTC are unchanged. Within the estimation process, the region and urban-rural dimensions of $y_{asrut}^{\text{True}}$ are aggregated away before the obesity counts are supplied to the data models.

The data model for the schools data is expanded to include $r$ and $u$ subscripts,

$$\log y_{asrut}^{\text{EstS}} \sim \text{N}(\log y_{asrut}^{\text{True}} + \gamma_a, \varsigma_y^2) \tag{15}$$

$$\gamma_a \sim \text{N}(0, \varsigma_\gamma^2). \tag{16}$$

The revised data model for schools, like the original data model for schools, only allows biases to vary by age. Biases may in fact vary by region or by urban-rural residence, but with only one source of data on subnational variation in obesity prevalence, there is limited scope for distinguishing between subnational variation in coverage and subnational variation in actual prevalence.

## Results

The top panel of Fig 5 shows results from Model 4. To save space, Fig 5 only includes results for females in urban areas; results for males and for rural areas are shown in the S1 File. Each

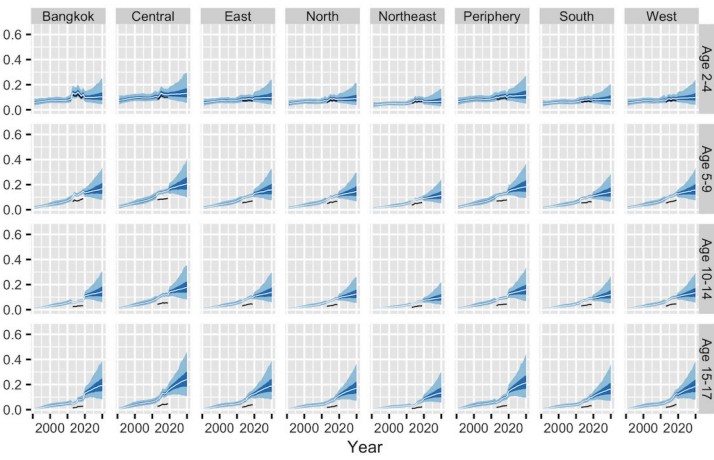

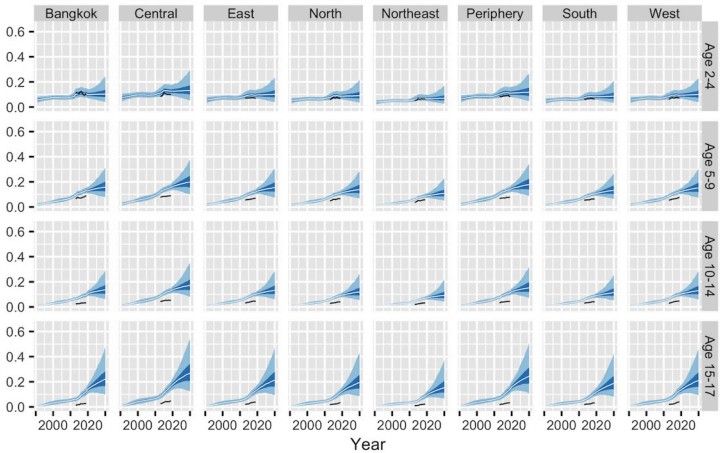

**Fig 5. Estimates and forecasts of obesity prevalence for females in urban areas, from models 4 and 5.** The top panel shows results from our first subnational model (Model 4), and the bottom panel shows results from our revised model (Model 5). The light bands represent 95% credible intervals, the dark bands represent 50% credible intervals, and the white lines represent medians. The black symbols represent direct school-based estimates from Fig 2.

age group in Fig 5 follows a similar trajectory to the corresponding national-level age group in Fig 3, though these trajectories vary slightly between regions, reflecting patterns in the raw data in Fig 2. The Northeast region, for instance, has relatively low prevalences in Fig 5, reflecting the relatively low direct estimates of prevalence shown in Fig 2. Model 4 seems, in many ways, to be giving sensible results.

One implausible feature of the results from Model 4, however, is the sudden departures from long-term trends that occur in many series in response to the schools data. In Bangkok, for instance, estimates for ages 2–4 and 5–9 shift sharply upwards during the years where the schools data are available, while estimates for ages 10–14 and 15–17 shift sharply downwards. In regions such as Bangkok and Central Thailand, the estimates closely track year-to-year changes in the schools data.

The sudden departures from long-term trends and the close tracking of the schools data evident in the top panel of Fig 5 reflect the fitted values for parameters $\sigma$ and $\varsigma_y$ in Model 4. Parameter $\sigma$, from the subnational equivalent of (2), governs the amount of idiosyncratic variation in true obesity prevalence. Parameter $\varsigma_y$, from (15), governs the accuracy of the schools data after accounting for age-specific biases. The point estimate for $\sigma$ is 0.132 in Model 4, compared with 0.013 in Model 3, while the point estimate for $\varsigma_y$ is 0.023 in Model 4, compared with 0.010 in Model 3. These numbers imply that subnational prevalence rates are much less stable than national rates, while subnational school-based measures of these rates are only slightly less accurate than national school-based measures.

We would, instead, expect Model 4 to have slightly higher values for $\sigma$ than Model 3, and much higher values for $\varsigma_y$. Some increase in idiosyncratic variation in prevalence rates when moving from the national level to the subnational level is plausible, but a 10-fold increase is not. Conversely, given the low coverage rates in regions such as Bangkok, and the complications introduced by student migration, subnational school-based estimates of obesity prevalence could be expected to be substantially less reliable than those national-level estimates. Some deficiency in Model 4 seems to be leading it to misrepresent variation in true prevalences and misrepresent the accuracy of the schools data.

## Model 5: Adjusting the data model for the schools data

### Methods

We obtain our final model, Model 5, by adjusting the data model for the schools data. In Model 4, and in earlier models, we use the same half-normal prior for $\varsigma_y$ that we use for scale parameters in the system model. A half-normal prior favours values near zero. In the system model, using a prior for scale parameters that favours values near zero damps down variation in parameters, such as the $\beta$ or $\pi$, that are governed by the scale parameter. Damping down variation in these parameters provides robustness to noise in the data. In the data model for schools, however, using a prior for $\varsigma_y$ that favours values near zero reduce robustness, instead of increasing it. Values of $\varsigma_y$ near zero imply that (after accounting for age-specific biases) most observed variation in schools data reflects real variation in underlying prevalences. When $\varsigma_y$ is low, variation in the data is propagated through to estimates of underlying prevalences. This is not appropriate behavior when, as with the subnational schools data, the data is likely to have substantial measurement error.

We replace the half-normal prior for $\varsigma_y$ with one that no longer favours values near zero. We switch from a half-normal prior to a scaled inverse-$\chi^2$ distribution, which has a mode away from zero and a long right tail. For mathematical convenience, we apply the prior to $\varsigma_y^2$ rather than to $\varsigma_y$ itself. A scaled inverse-$\chi^2$ distribution has a degrees-of-freedom parameter which, roughly speaking, controls the dispersion of the distribution, and a scale-squared

parameter which controls the mean. We derive values for these parameters that try to reflect the plausible range of values for $\varsigma_y^2$.

Let $p_{asrut}$ be a direct estimate of obesity prevalence from the schools data, as depicted in Fig 2. Our data model for the schools data implies that

$$\log p_{asrut} = \log \pi_{asrut} + \gamma_a + u_{asrut}. \tag{17}$$

where $\pi_{asrut}$ is the true prevalence, and $u_{asrut}$ has a normal distribution with mean 0 and variance $\varsigma_y^2$. We approximate $\log \pi_{asrut}$ with $z_{asrut}\eta$, where $\eta$ contains the same main effects and interactions as our subnational system model and $v_{asrut}$ is a vector of 1s and 0s. Substituting into (17) gives

$$\log p_{asrut} = z_{asrut}\eta + \gamma_a + u_{asrut} \tag{18}$$

$$= z'_{asrut}\eta' + u_{asrut}, \tag{19}$$

where $z'_{asrut}\eta' == z_{asrut}\eta + \gamma_a$. We fit the model of (19) using least squares, as implemented in function `lm` in *R* package **stats**, obtaining a point estimate for $\varsigma_y^2$ of 0.016.

We set the degrees-of-freedom parameter for the scaled inverse-$\chi^2$ prior distribution to 30 and the scale-squared parameter to 0.015. These values imply that there is a 95% chance that the true value of $\varsigma_y^2$ is between 0.010 and 0.027.

## Results

Results from our revised model, Model 5, for females in urban areas, are shown in the lower panel of Fig 5. Results for males and for rural areas are included in the S1 File. The estimate from Model 5 are smoother than those from Model 4. Estimated prevalences still respond to changes in the schools data, but not nearly as closely, and without the dramatic upward and downward shifts.

Replacing the prior for $\varsigma_y$ leads, as expected, to lower estimates for $\sigma$ and higher estimates for $\varsigma_y$. The new point estimate for $\sigma$ is 0.073, compared to 0.132 in Model 4, and the new point estimate for $\varsigma_y$ is 0.105, compared to 0.023 in Model 4.

## Checking the model

An essential part of any modelling workflow is to assess how sensitive the results are to alternative possible specifications, and to consider whether the model has captured all the substantively-important features of the system under study [32]. A full suite of model checking would require more space than is available here, but we present two illustrative examples.

### Sensitivity to priors for time main effects and interactions

As discussed above, we suspect that the WHO country-level estimates understate annual variability in obesity prevalence, but are unsure by how much. In Models 3–5, we use priors that, roughly speaking, entail twice as much annual variability than is implied by the original WHO estimates. Here we investigate how the results vary with alternative choices. To gain insights into the way that the priors interact with other parts of the model, we show results for each of the broader model classes considered in the paper.

Each panel of Fig 6 shows national-level estimates and forecasts, aggregating over sex and age. Each column of panels shows how result vary with the choice of prior. The priors are ordered from weakest to strongest. The first column is our original weakly-informative half-normal prior. The fourth column is the prior obtained by using the unadjusted values obtained

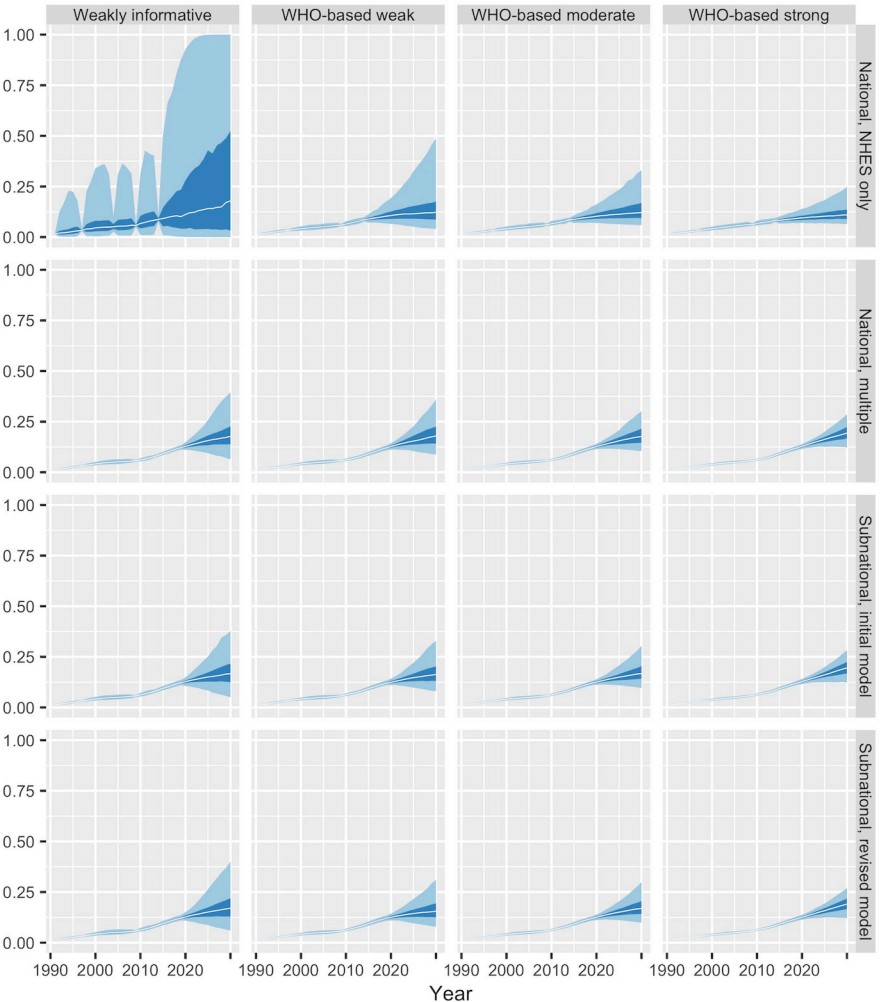

**Fig 6. Comparison of national-level estimates and forecasts of obesity prevalence from different combinations of model type and priors for variance terms in time main effects and interactions.** The estimates and forecasts aggregate over age, sex, region, and urban-rural residence. Each row of panels represents one model type, and each column represents one prior.

from the WHO model. The third column is the prior obtained by inflating variances by a factor of 2. The second column is the prior obtained by inflating variances by a factor of 4. Each row of the figure shows results for a different overall model specification, starting with the national-level NHES-only model in row 1, and finishing with the subnational model with the revised data model for schools in row 4. Panel (1, 1) in the figure corresponds to Model 1; panel (2, 1) to Model 2; panel (2, 3) to Model 3; panel(3, 3) to Model 4; and panel (4, 3) to Model 5.

Replacing the default weakly informative time prior with a WHO-based prior reduces forecast uncertainty under all four model specifications, though it has the biggest effect in the national-level NHES-only model. With the exception of the national-level NHES-only model, stronger WHO-based priors lead to more linear forecasts, with less tapering-off in growth rates towards the end of the period. When comparing across WHO-based priors, however, the choice of prior has only a minor effect on the shape or uncertainty of estimates in years where there is data.

## Replicate data checks

A Bayesian statistical model treats the observed data as a draw from a probability distribution. If the model is a good representation of the process being studied, then it should be possible to use the model to randomly generate hypothetical datasets that look similar to the real dataset. Conversely, if hypothetical datasets generated by the model are systematically different from the real dataset, then the model may be deficient. Bayesians call the technique of comparing hypothetical datasets with the real dataset "replicate data checks" [28, 32].

Replicate data checks should be targeted at possible weak points in the model. One possible weak point of Model 5 is that the prior model for the prevalences $\pi_{asrut}$ assumes a common time trend across all regions, and across urban and rural areas. Any geographical variation in the pace at which measured obesity is increasing is assumed to arise from random variation in probabilities of being obese, random variation in obesity counts given probabilities, and random variation in measured counts given true counts. It is possible that this approach implies too much uniformity across regions and across urban and rural areas.

We use replicate data checks to assess the ability of Model 5 to generate geographical variation in rates of change. We focus on the schools dataset, since the schools dataset is the only one that is disaggregated by geography. Each replicate dataset is generated as follows:

1. Randomly select a draw $k$ from the sample from the joint posterior distribution from Model 5, and obtain values $\beta^{(k)}$, $\sigma^{(k)}$, $\gamma_a^{(k)}$ and $\varsigma_y^{(k)}$.

2. Plug $\beta^{(k)}$ and $\sigma^{(k)}$ into the prior model for prevalences, and generate values $\pi_{asrut}^{(k)}$.

3. Plug the $\pi_{asrut}^{(k)}$ into (14), and generate values $y_{asrut}^{(k)}$ for true numbers of obese children.

4. Plug $y_{asrut}^{(k)}$ into (15), and generate values $y_{asrut}^{\text{EstS}(k)}$ for school-based estimates of obese children.

Repeating this process $K$ types yields $K$ replicate datasets. The observed and replicate datasets are too big to work with directly, so we calculate summary measures of geographical variation in rates of change for each dataset and compare the summary measures instead. For each combination of age, sex, region, and urban-rural residence in each dataset, we fit a straight line through values for $y_{asrut}^{\text{EstS}}$ versus time. The slopes of these lines are our summary measures.

Fig 7 shows the results from these calculations. The first column of panels gives results for the observed dataset, and the remaining columns give results for the replicate datasets. Each dot represents a slope estimate. To save space, the figure only shows results for females.

What we are looking for in Fig 7 is evidence of systematic differences between the observed dataset and the replicate datasets. Given that all datasets, including the observed one, have an element of randomness, we are not looking for complete agreement, but instead for whether the observed dataset is in some sense an outlier.

Inspection of Fig 7 suggests that the observed dataset is not an outlier. The observed dataset contains about the same amount of geographical variation in rates of change as the replicate datasets do. The use of a common time trend across regions and urban-rural residents appears to be a reasonable approximation.

## Discussion

In this paper, we have developed disaggregated estimates and forecasts of obesity among Thai children. We have found that trends differ substantially by age: prevalences for children under 5 have shown little change since the early 1990s, while prevalences for children aged 15–17 have, in recent years, increased sharply, with a strong prospect of reaching rates of 25% or more by 2030. Prevalences have varied across different parts of the country, though forecasted

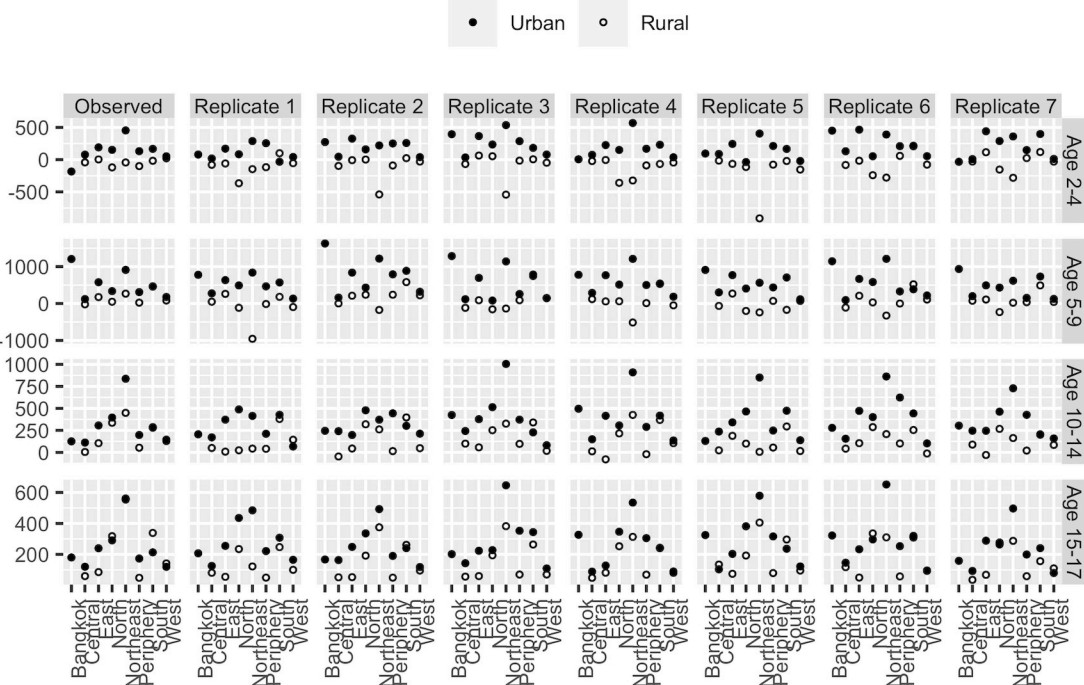

**Fig 7. Using replicate datasets to assess the ability of the model to capture geographical variability in rates of change for obesity.** Each columns shows results from one dataset. Each dot represents the slope from a regression of numbers of obese children against time. The figure shows results for females only.

prevalences show substantial overlap. Supplementing the National Health Examination Survey data with additional survey and administrative data reduces uncertainty considerably, and exploiting international data on trends in obesity prevalence reduces uncertainty further. Even with the extra information, however, the forecasts are still sensitive to alternative assumptions about the variability in rates of change. Our ability to measure geographical variation is also constrained by the fact that our only source of information on geographical variation is subject to measurement error. Further improvements in the estimates and forecasts are likely to require additional data.

The apparent divergence in trends for children below and above age 5 is new to the study of obesity in Thailand. Most children in Thailand begin attending some form of schooling around this age, which suggests that there may be something about the school environment that leads to higher rates of obesity. In addition, the forecasts that obesity will become common across all regions of the country, and in both urban and rural areas, suggests that policies to address rising obesity rates need to be implemented nationally, and not just in places where obesity is already high.

The analytical framework used in this paper can accommodate a wide range of applications involving disaggregated estimation and forecasting. The system model, the set of unobserved counts, and the data models can all be customised to the problem at hand. The resulting models can be complex. However, the individual components from which the models are composed are often simple and intuitive, and the models can be built up piece by piece. Combining the assessment of data quality, the estimation of underlying rates, and the forecasting of future values into a single framework allows us to capture uncertainties in a unified and internally-consistent way.

The ability to combine multiple data sources allows frequent updating of estimates and forecast. In the case of Thai obesity, for instance, estimates and forecasts of obesity can be revised each time new schools data becomes available, rather than waiting for a new round of the National Health and Examination Survey—though regular updates of the NHES data are still important to adjust for errors in the schools data.

A dictinctively Bayesian part of our analysis is the use of informative prior distributions. We use an informative prior distribution to describe the plausible range for annual change in obesity rates, and use an informative prior distribution to describe the likely size of measurement errors in the schools data. In both cases, the informative prior distributions are grounded in quantitative analyses, and are transparent and replicable. Incorporating informative priors into the model increases the plausibility and precision of the resulting estimates and forecasts.

## Supporting information

**S1 File. This file contains all the supporting figures and text.**
(PDF)

## Acknowledgments

We would like to thank the National Health Examination Survey, the Office of the Basic Education Commission of Thailand, the Holistic Development of Thai Children Project, the National Statistics Office of Thailand, the Thai National Economic and Social Development Board, and the World Health Organization for data used in this paper. We are also grateful to colleagues who provided feedback on earlier versions of the paper.

## Author Contributions

**Conceptualization:** Jongjit Rittirong.

**Data curation:** John Bryant, Jongjit Rittirong, Wichai Aekplakorn, Ladda Mo-suwan, Pimolpan Nitnara.

**Formal analysis:** John Bryant.

**Funding acquisition:** Jongjit Rittirong.

**Methodology:** John Bryant, Jongjit Rittirong, Wichai Aekplakorn, Ladda Mo-suwan, Pimolpan Nitnara.

**Project administration:** Jongjit Rittirong.

**Resources:** Jongjit Rittirong.

**Software:** John Bryant.

**Supervision:** Jongjit Rittirong.

**Validation:** John Bryant, Jongjit Rittirong, Wichai Aekplakorn, Ladda Mo-suwan, Pimolpan Nitnara.

**Visualization:** John Bryant.

**Writing – original draft:** John Bryant.

**Writing – review & editing:** John Bryant, Jongjit Rittirong, Wichai Aekplakorn, Ladda Mo-suwan, Pimolpan Nitnara.

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
