## [Decision Letter · Decision Letter 0]

26 Oct 2021

PONE-D-21-23949A Bayesian approach to combining multiple information sources: Estimating and forecasting childhood obesity in ThailandPLOS ONE

Dear Dr. Rittirong,

Thank you for submitting your manuscript to PLOS ONE. After careful consideration, we feel that it has merit but does not fully meet PLOS ONE’s publication criteria as it currently stands. Therefore, we invite you to submit a revised version of the manuscript that addresses the points raised during the review process.

We look forward to receiving your revised manuscript.

Kind regards,

Mona Pathak, PhD

Academic Editor

PLOS ONE

Journal Requirements:

Reviewers' comments:

Reviewer's Responses to Questions

**Comments to the Author**

1. Is the manuscript technically sound, and do the data support the conclusions?

Reviewer #1: Yes

Reviewer #3: Yes

2. Has the statistical analysis been performed appropriately and rigorously? 

Reviewer #1: Yes

Reviewer #3: Yes

3. Have the authors made all data underlying the findings in their manuscript fully available?

Reviewer #1: Yes

Reviewer #3: Yes

4. Is the manuscript presented in an intelligible fashion and written in standard English?

Reviewer #1: Yes

Reviewer #3: Yes

5. Review Comments to the Author

Reviewer #1: I think that the paper is well written and the gradual build-up, from simple to more complex models, makes it easy to follow authors' presentation and to see how different information that are combined from various sources in ultimately constructing a model to analyze and predict children obesity in Thailand. I am listing here a few concerns that I hope perhaps the authors can clarify in the paper in order for a better understanding from the users.

(1) after the first model, information from HDTC and Schools data are used to strengthen inference. Instead of using observed data, you estimated the truth population total (n's) and true obese children (y's). They become the data models. Once you have three models, you combined them in a single analysis. Can you be more specific on how this combined analysis is carried out? My confusion, perhaps a misunderstanding, is that if you estimate three true population totals (n^TRUE's) and three y^TRUE, and if you naively combine them, estimation would shrink the standard errors because you over count the population total: p(1-p)/n if n gets to be artificially large (3xn^TURE).

(2) model 1, equation 4-6, page 8. That time-dependent 'damped linear trend' model has its advantages in offering a way to model how the time effects change. But it seems to give you some trouble, and estimation is sensitive to the variance parameters, e.g. tao_delta^2, and later you pointed out that, when more information is used, you ended up gaining a better estimate of these parameters, which help in stabilizing the estimates, etc. I feel that there are other practical ways of modeling time-varying beta_t coefficients jointly, without having to resort to estimating the random drifts. For example, multivariate normal prior with certain temporal covariance type. Those probably can also be more computationally stable, and less sensitive to the choices of the prior. Have you considered using alternatives?

(3) Why some data models you use normal likelihood (e.g. Eq 9), others you take log transformation (Eq 10)?

(3a) Your data models depend on externally-estimated parameter, such as kappa_ast in model of y^EstN_ast. Can you provide a bit more explanations, at least intuitively, why that is a reasonable idea?

(3b) you indicate that, while principally it is possible to estimate all systematic differences in data model, you end up choosing only to include age-specific gamma_alpha, and explain that having too many such bias parameters would make the estimation more difficult. To me, that is an indicate that not enough information in the data can be used to estimate such bias parameters (theoretically, but practically need more data). This leads to a question that perhaps you can address in the paper: is it then even necessary to include the gamma_alpha parameter? Will your model fit be just as good if that parameter is not there? Maybe this amounts to doing some sensitivity analysis in comparing models with or without that parameter and see if any significant different conclusions can be observed.

(4) minor point: page 3, line 61, what are bayescomb and bayescombwho?

Reviewer #3: I thank the authors for investigating such an important subject; I have some minor comments:

- Method and Results should be written in separate sections.

- Line 53-55: the third extension is missing.

- Line 173-180: Authors assigned Normal-prior to variables Sex and Age. Considering the positive nature of the indicator variable, why did not the authors use the half-normal prior for these variables?

6. PLOS authors have the option to publish the peer review history of their article (what does this mean?). If published, this will include your full peer review and any attached files.

Reviewer #1: No

Reviewer #3: No

---

## [Author Response · Author response to Decision Letter 0]

4 Dec 2021

Journal Requirements:

Response: We have reformatted the manuscript to meet PLOS ONE's style requirements, including figure captions. 

Response: Data and code for the WHO-based prior distributions are at https://github.com/johnrbryant/bayescombwho and data and code for the rest of the paper are at https://github.com/johnrbryant/bayescomb

Response: This study does not involve primary data and used only secondary datasets from various sources. The ethics statement (Code of Exemption) is included in the Data section. 

Ethics statement

This study was approved by the Institute for Population and Social Research 107

Institutional Review Board (IPSR-IRB), at Mahidol University, Thailand (COE. 108

No. 2019/07-278).

Response: We included captions in the Supporting Information file, and have updated all in-text citations.

Response: We have reviewed all references to ensure that they have not been retracted. The reference format was checked and completed. 

Reviewers' comments:

Reviewer's Responses to Questions

Comments to the Author

1. Is the manuscript technically sound, and do the data support the conclusions?

Reviewer #1: Yes

Reviewer #3: Yes

Response: Thank you

2. Has the statistical analysis been performed appropriately and rigorously?

Reviewer #1: Yes

Reviewer #3: Yes

Response: Thank you

3. Have the authors made all data underlying the findings in their manuscript fully available?

Reviewer #1: Yes

Reviewer #3: Yes

Response: Thank you

4. Is the manuscript presented in an intelligible fashion and written in standard English?

Reviewer #1: Yes

Reviewer #3: Yes

Response: Thank you

5. Review Comments to the Author

Response: The explanation responds to each comments shown above. 

Reviewer #1: I think that the paper is well written and the gradual build-up, from simple to more complex models, makes it easy to follow authors' presentation and to see how different information that are combined from various sources in ultimately constructing a model to analyze and predict children obesity in Thailand. I am listing here a few concerns that I hope perhaps the authors can clarify in the paper in order for a better understanding from the users.

Response: Thank you

(1) after the first model, information from HDTC and Schools data are used to strengthen inference. Instead of using observed data, you estimated the truth population total (n's) and true obese children (y's). They become the data models. Once you have three models, you combined them in a single analysis. Can you be more specific on how this combined analysis is carried out? My confusion, perhaps a misunderstanding, is that if you estimate three true population totals (n^TRUE's) and three y^TRUE, and if you naively combine them, estimation would shrink the standard errors because you over count the population total: p(1-p)/n if n gets to be artificially large (3xn^TURE).

Response: This is an interesting point, and we suspect that the same question may occur to other readers. We have added the following paragraph to methodological description of the first model (page 13, line 322):

“The likelihood from combining the three datasets has the form p(data|true obesity counts) = p(NHES data|true obesity counts) × p(HDTC data|true obesity counts) × p(Schools data|true obesity counts). 

The (unobserved) true obesity counts appear multiple times in the likelihood. The repetition does not, however, cause any problems. It is analogous to having the same regression coefficients occur multiple times in the likelihood for a regression model.”

(2) model 1, equation 4-6, page 8. That time-dependent 'damped linear trend' model has its advantages in offering a way to model how the time effects change. But it seems to give you some trouble, and estimation is sensitive to the variance parameters, e.g. tao_delta^2, and later you pointed out that, when more information is used, you ended up gaining a better estimate of these parameters, which help in stabilizing the estimates, etc. I feel that there are other practical ways of modeling time-varying beta_t coefficients jointly, without having to resort to estimating the random drifts. For example, multivariate normal prior with certain temporal covariance type. Those probably can also be more computationally stable, and less sensitive to the choices of the prior. Have you considered using alternatives?

Response: It is true that a less flexible prior for the time effects would be easier to estimate, especially given our limited number of observations. However, it seems that we need all of the features of the damped linear trend, in order to capture the main features of the data. For instance, it would be much simpler if we could assume that rates of change were constant over time, but the evidence suggest that rates of change for teenagers have in fact increased since 2010. The software we are using does not give a wide choice of priors for time effects. However, we suspect that other priors with similar levels of flexibility would lead to similar difficulties.

We have added the following paragraph to our discussion of the damped linear trend prior (page 9, line 202-207):

“As we discuss below, the flexibility of the damped linear trend prior makes it challenging to fit. However, less flexible versions of the prior could potentially miss important features of the data. It is, for instance, tempting to assume that, within each combination of age and sex, rates of change are constant over time. Doing so would, however, reduce our ability to detect turning points, and could produce forecasts that were inappropriately confident.”

(3) Why some data models you use normal likelihood (e.g. Eq 9), others you take log transformation (Eq 10)?

Response: We have completely rewritten the paragraphs describing Eq 9. The new paragraphs are as follows (page 12, line 288-298): 

“To construct our data model for the NHES, we rely on features of the design of the survey, which is a common strategy in Bayesian analyses of multiple data sources [8,15]. The design of the survey implies that the NHES estimates should be unbiased, and that errors in these estimates should be approximately normally distributed. Moreover, the standard deviations for these errors can be estimated through design-based methods 

that exploit information about the survey including sample weights.

(3a) Your data models depend on externally-estimated parameter, such as kappa_ast in model of y^EstN_ast. Can you provide a bit more explanations, at least intuitively, why that is a reasonable idea?

Response: Our data model for the NHES is 

yEstN ∼N(yTrue,κ2 ) (9) 

where the superscript ‘EstN’ denotes ‘estimates derived from the NHES’. We set the κast equal to the standard deviations that the R package survey produces alongside the estimates yTrue. (Code for the design-based calculations s included in the repository https://github.com/johnrbryant/bayescomb.)”

In addition, to motivate the use of log units rather than natural units in the schools data model, we have added the following sentence (page 13, line 310-311):

“The use of logs implies that the data model is expressed in terms of percentage errors, rather than absolute errors.”

(3b) you indicate that, while principally it is possible to estimate all systematic differences in data model, you end up choosing only to include age-specific gamma_alpha, and explain that having too many such bias parameters would make the estimation more difficult. To me, that is an indicate that not enough information in the data can be used to estimate such bias parameters (theoretically, but practically need more data). This leads to a question that perhaps you can address in the paper: is it then even necessary to include the gamma_alpha parameter? Will your model fit be just as good if that parameter is not there? Maybe this amounts to doing some sensitivity analysis in comparing models with or without that parameter and see if any significant different conclusions can be observed.

Response: We have added a sensitivity test where, as suggested, we drop the gamma_alpha term. The results, shown in section “Dropping the age-specific bias term from the data model for schools” in the Supplementary Materials.The gamma_alpha term is in fact needed. In the main section of the paper we write (Supplementary Material, page 2, line 18-28)

“Dropping the age-specific bias term from the data model for schools

Fig 2 below shows national estimates and forecasts when the data model for schools does not have an age-specific bias term. Fig 3b in the main text shows the equivalent estimates and forecasts when the data model for schools does have an age-specific bias term. Without the bias term, there is one-off change in apparent prevalences for children aged 2–4 when moving from the period covered by the NHES and HDTC data to the period covered by the schools data. The widths of the credible intervals are also reduced. The one-off change appears implausible. Given that there is uncertainty about age-specific biases, the wider credible intervals of the main model are, we believe, more appropriate.”

(4) minor point: page 3, line 61, what are bayescomb and bayescombwho?

Response: These were supposed to be hyperlinks to GitHub repositories, but did not print correctly. We have replaced these links.

Reviewer #3: I thank the authors for investigating such an important subject; I have some minor comments:

- Method and Results should be written in separate sections.

Response: We have added “Methods” and “Results” subheadings to the descriptions of Models 1, 2, 3, 4, 5. 

- Line 53-55: the third extension is missing.

Response: Thank you for alerting us to the missing third extension! We have renumbered the extensions.

- Line 173-180: Authors assigned Normal-prior to variables Sex and Age. Considering the positive nature of the indicator variable, why did not the authors use the half-normal prior for these variables?

Response: We have added a sentence (page 7, line 169), “Transformation to the logit scale implies that values are no longer bounded by 0 and 1.” 

We illustrate the implications for the sex prior with the sentence (page 10, line 180) “This prior captures the idea that, on a logit scale, we might see female-male differences of -0.1 or 1.2, for instance, but not -10 or 120.” (Previously we used the numbers 0.1 and 10, rather than -0.1 and -10.)

6. PLOS authors have the option to publish the peer review history of their article (what does this mean?). If published, this will include your full peer review and any attached files.

Do you want your identity to be public for this peer review? For information about this choice, including consent withdrawal, please see our Privacy Policy.

Reviewer #1: No

Reviewer #3: No

Response: Sorry, we made a mistake. This should be revised to be “yes”. All authors have agreed to reveal their identity including name, title, and workplace.

---

## [Decision Letter · Decision Letter 1]

16 Dec 2021

A Bayesian approach to combining multiple information sources: Estimating and forecasting childhood obesity in Thailand

PONE-D-21-23949R1

Dear Dr. Rittirong,

We’re pleased to inform you that your manuscript has been judged scientifically suitable for publication and will be formally accepted for publication once it meets all outstanding technical requirements.

Kind regards,

Fabio Rapallo, Ph.D.

Academic Editor

PLOS ONE

Additional Editor Comments (optional):

Reviewers' comments:

Reviewer's Responses to Questions

**Comments to the Author**

1. If the authors have adequately addressed your comments raised in a previous round of review and you feel that this manuscript is now acceptable for publication, you may indicate that here to bypass the “Comments to the Author” section, enter your conflict of interest statement in the “Confidential to Editor” section, and submit your "Accept" recommendation.

Reviewer #1: All comments have been addressed

Reviewer #3: All comments have been addressed

2. Is the manuscript technically sound, and do the data support the conclusions?

Reviewer #1: Yes

Reviewer #3: Yes

3. Has the statistical analysis been performed appropriately and rigorously? 

Reviewer #1: Yes

Reviewer #3: Yes

4. Have the authors made all data underlying the findings in their manuscript fully available?

Reviewer #1: Yes

Reviewer #3: Yes

5. Is the manuscript presented in an intelligible fashion and written in standard English?

Reviewer #1: Yes

Reviewer #3: Yes

6. Review Comments to the Author

Reviewer #1: Thanks for taking the time to answer all of my questions and address concerns. I have no further comments.

Reviewer #3: (No Response)

7. PLOS authors have the option to publish the peer review history of their article (what does this mean?). If published, this will include your full peer review and any attached files.

Reviewer #1: No

Reviewer #3: No

---

## [Editor Report · Acceptance letter]

21 Dec 2021

PONE-D-21-23949R1 

A Bayesian approach to combining multiple information sources: Estimating and forecasting childhood obesity in Thailand 

Dear Dr. Rittirong:

I'm pleased to inform you that your manuscript has been deemed suitable for publication in PLOS ONE. Congratulations! Your manuscript is now with our production department. 

Kind regards, 

on behalf of

Dr. Fabio Rapallo 

Academic Editor

PLOS ONE